# Development of an enhanced scoring system to predict ICU readmission or in-hospital death within 24 hours using routine patient data from two NHS Foundation Trusts

Marco A F Pimentel,[1] Alistair Johnson,[2] Julie Lorraine Darbyshire [iD],[3] Lionel Tarassenko,[1] David A Clifton,[1] Andrew Walden,[4] Ian Rechner,[4] Peter J Watkinson [iD],[5] J Duncan Young[5]

¹Department of Engineering Science, University of Oxford, Oxford, UK
²Institute of Medical Engineering & Science, Massachusetts Institute of Technology, Cambridge, Massachusetts, USA
³Department of Primary Care Health Sciences, Oxford University, Oxford, UK
⁴Royal Berkshire NHS Foundation Trust, Reading, UK
⁵Nuffield Department of Clinical Neurosciences, University of Oxford, Oxford, UK

**Correspondence to**
Dr Julie Lorraine Darbyshire;
julie.darbyshire@phc.ox.ac.uk

## ABSTRACT

**Rationale** Intensive care units (ICUs) admit the most severely ill patients. Once these patients are discharged from the ICU to a step-down ward, they continue to have their vital signs monitored by nursing staff, with Early Warning Score (EWS) systems being used to identify those at risk of deterioration.

**Objectives** We report the development and validation of an enhanced continuous scoring system for predicting adverse events, which combines vital signs measured routinely on acute care wards (as used by most EWS systems) with a risk score of a future adverse event calculated on discharge from the ICU.

**Design** A modified Delphi process identified candidate variables commonly available in electronic records as the basis for a 'static' score of the patient's condition immediately after discharge from the ICU. L1-regularised logistic regression was used to estimate the in-hospital risk of future adverse event. We then constructed a model of physiological normality using vital sign data from the day of hospital discharge. This is combined with the static score and used continuously to quantify and update the patient's risk of deterioration throughout their hospital stay.

**Setting** Data from two National Health Service Foundation Trusts (UK) were used to develop and (externally) validate the model.

**Participants** A total of 12 394 vital sign measurements were acquired from 273 patients after ICU discharge for the development set, and 4831 from 136 patients in the validation cohort.

**Results** Outcome validation of our model yielded an area under the receiver operating characteristic curve of 0.724 for predicting ICU readmission or in-hospital death within 24 hours. It showed an improved performance with respect to other competitive risk scoring systems, including the National EWS (0.653).

**Conclusions** We showed that a scoring system incorporating data from a patient's stay in the ICU has better performance than commonly used EWS systems based on vital signs alone.

## STRENGTHS AND LIMITATIONS OF THIS STUDY

⇒ Over 17 000 vital sign measurements from >400 patients were included in the cohort.
⇒ Variable selection was informed by systematic review and Delphi process prior to data extraction.
⇒ Data extraction variables were limited to those available.
⇒ Performance was assessed using common methodology for Early Warning Score systems.
⇒ Compound outcome (including in-hospital mortality and intensive care unit readmission) limits prediction of inevitable death and improves prediction of preventable deaths.

**Trial registration number** ISRCTN32008295.

## INTRODUCTION

Patients hospitalised for acute conditions can suffer adverse events such as cardiac arrests or unplanned admission to a higher-acuity area. These events are usually preceded by changes in vital signs some hours before.[1–5] Early detection of these changes might prevent some of the subsequent adverse events, so many simple Early Warning Score (EWS) systems have been developed and deployed clinically.[6–8] EWS systems are now recommended to be used as part of routine care by the UK National Institute for Health and Clinical Excellence[9–11] and are widely used in other healthcare settings.

EWS systems initially used purpose-designed paper charts.[12 13] An increasing number of healthcare providers now use electronic systems[14 15] to gather physiological data and calculate scores.[16] With the increasing use

of electronic patient records, there is scope to develop improved EWS systems that include many more variables than are currently used, potentially leading to increasing predictive accuracy by using more complex algorithms to calculate scores.[17]

We hypothesise that an EWS system could better predict adverse events if it included variables from a patient's electronic medical record. As a proof of concept, we studied patients discharged to acute care wards from intensive care units (ICUs). These patients have a detailed electronic patient record containing data about their ICU stay, and typically having a high rate of adverse events occurring after discharge to a ward. In addition, there is a large body of literature describing variables that are correlated with adverse events in these patients.

This study describes the development of an enhanced scoring system for predicting adverse events in patients discharged from ICUs which combines (1) routine vital signs measured on acute care wards (as used by most EWS systems) with (2) a risk score of a future adverse event calculated on discharge from the ICU. The study further validates the ability of the proposed scoring system and other published EWS systems to identify patients at risk of death or readmission to the ICU.

## MATERIALS AND METHODS

This study is reported in line with the Transparent Reporting of a multivariable prediction model for Individual Prognosis or Diagnosis statement.[18]

### Data sources

Data from two National Health Service (NHS) Foundation Trusts (UK) were used in this study. The Oxford University Hospitals NHS Foundation Trust (OUH) has two adult general ICUs. The Royal Berkshire Hospital NHS Foundation Trust (RBH) in Reading, UK has a single adult general ICU.

We used routinely collected data, stored in the ICU computerised information systems (CIS). The OUH and RBH CIS contain all measurements of physiological status and other relevant clinical information, such as patient demographics; details of treatments and interventions; and laboratory test results recorded during a patient's ICU stay. An average of 655 data items are recorded daily for each patient. All the ICUs in this study use a Philips Healthcare CIS (Philips Healthcare, Eindhoven, the Netherlands). In addition, we used the linked datasets submitted to the UK national comparative audit for ICUs or Intensive Care National Audit & Research Centre Case Mix Programme.[19]

During the study period, neither organisation had electronic recording of vital signs for patients in acute care wards outside the ICUs. Vital sign data were collected prospectively for recruited patients in both organisations for the first 14 days after discharge from the ICU. This timeline was chosen because the average duration of hospital stay after discharge from intensive care is

around 10 days in the UK (https://www.icnarc.org/Our-Audit/Audits/Cmp/Reports/Summary-Statistics). The physiological data from each patient's paper observation chart were entered into an electronic database. To verify the data entry process, 55% of patients' data were double-entered.

All analyses were conducted on anonymised data.

### Participants

All completed adult admissions to OUH ICUs from June 2006 to December 2015 and to the RBH ICU from April 2010 to December 2015 were used to develop the ICU discharge score. Admissions were only considered when (1) patients were discharged alive from the ICU and (2) the discharge status of the patient episode (or hospital admission) was known (or recorded). We excluded patients discharged from the ICU for palliative care or transferred to another organisation. Vital sign data on wards were acquired prospectively after discharge from the ICU for a subset of patients who gave consent for this in OUH between April 2013 and December 2014, and in RBH from October 2013 and December 2014.

After exclusion criteria were applied, we split the database into 'development' and 'validation' sets by organisation (see figure 1), such that (1) we developed and evaluated the model using data from all valid admissions within the OUH group (8162 admissions) using cross-validation methods, and (2) we performed an external validation of the model using data from all valid admissions within the RBH group (3421 admissions).

Figure 1 shows the application of inclusion/exclusion criteria to patient admissions in order to derive the final cohorts in both organisations included in this study.

### Outcomes

The primary outcome was the first occurrence of either in-hospital death or readmission to the ICU. For the evaluation of EWS systems, we considered the compound outcome of in-hospital death or ICU readmission within the next N hours of a vital sign observation, in line with previous studies.[20 21] For the primary outcome, we set N=24 hours. We have also evaluated the systems for different values of N (as detailed below). Secondary outcomes were in-hospital death or readmission to the ICU, individually.

### Predictors

We used a conceptual model in which we estimated the risk of a patient experiencing an adverse event at the point of discharge from the ICU from variables recorded during their stay in the ICU. After discharge from the ICU, the patient's risk of experiencing an adverse event within the following 24 hours was calculated using both ward-recorded vital signs and the risk calculated at ICU discharge.

#### ICU-based feature representation

We used an evidence-based technique to select candidate variables to calculate the risk of deterioration at ICU

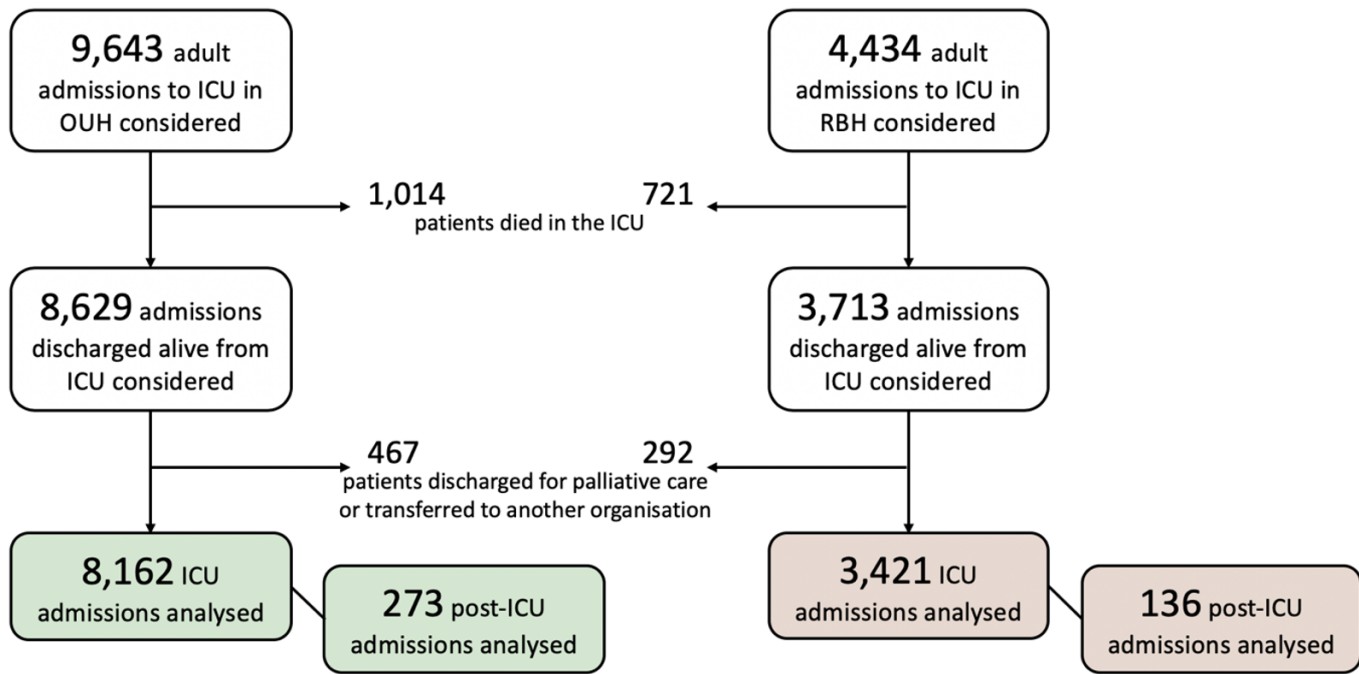

**Figure 1** Diagram showing the construction of the study cohorts from both organisations (OUH and RBH) included in the study by applying inclusion/exclusion criteria. ICU, intensive care unit; NHS, National Health Service; OUH, Oxford University Hospitals NHS Foundation Trust; RBH, Royal Berkshire Hospital NHS Foundation Trust.

discharge. We systematically reviewed studies reporting a significant (p<0.05) association between a variable recorded during an ICU stay and either in-hospital death or ICU readmission. The resulting list of candidate variables was reviewed by a panel of five clinical experts in a modified Delphi process who added other variables they expected to be predictive of adverse events. Candidate variables were then limited to those available in our electronic databases. These were either 'static' variables (mainly based on demographic information) or time-varying variables recorded repeatedly throughout the patient's stay in the ICU (see figure 2). To determine the risk of future compound outcome after discharge from the ICU, we derived a total of 161 candidate features from all candidate variables, which were then used for building a prediction model.

### Post-ICU feature representation

All vital sign observations recorded for 14 days after discharge from the ICU in the post-ICU subgroup were collected manually from patient records. Each set of vital signs includes heart rate, systolic blood pressure, respiratory rate, body temperature, neurological status assessment using the Alert-Verbal-Painful-Unresponsive Scale, peripheral oxygen saturation ($SpO_2$), a record of whether the patient was receiving supplemental oxygen at the time of the $SpO_2$ measurement, and the date and time of the observation. Vital sign measurements are typically recorded every 4 or 6 hours throughout the patient's stay on the ward (see figure 2). This was not an interventional study so conventional study size calculations are not appropriate. We have previously successfully developed

or validated novelty detection algorithms on samples of 150–200 patients.

The final list of candidate variables and features, and procedures for preprocessing (including dealing with missing data) are further described (see online supplemental file 1).

### Model development

To develop the risk scoring system, our approach assumes that at each vital sign observation performed after discharge from the ICU, the patient's current condition can be characterised (or represented) by a single risk estimate. Immediately after ICU discharge, the patient is assigned a risk score ($RS_1$) estimated from an ICU-based set of features, which is then updated using the abnormalities in their vital signs recorded during subsequent ward care ($RS_2$).

We therefore built the first model, $RS_1$, using the features derived from the variables acquired during the patients' stay in the ICU using the development set. We used an L1-regularised logistic regressor for predicting the compound outcome. For the second scoring system, $RS_2$, we applied a one-class classification method[22] using the vital sign data recorded after discharge from the ICU, as described previously.[23 24] Further details of the model development are available (see online supplemental file 1).

An overall risk score, the Risk Score Index (RSI), was subsequently determined using a simple time-dependent linear combination of the two constituent risk scores, such that:

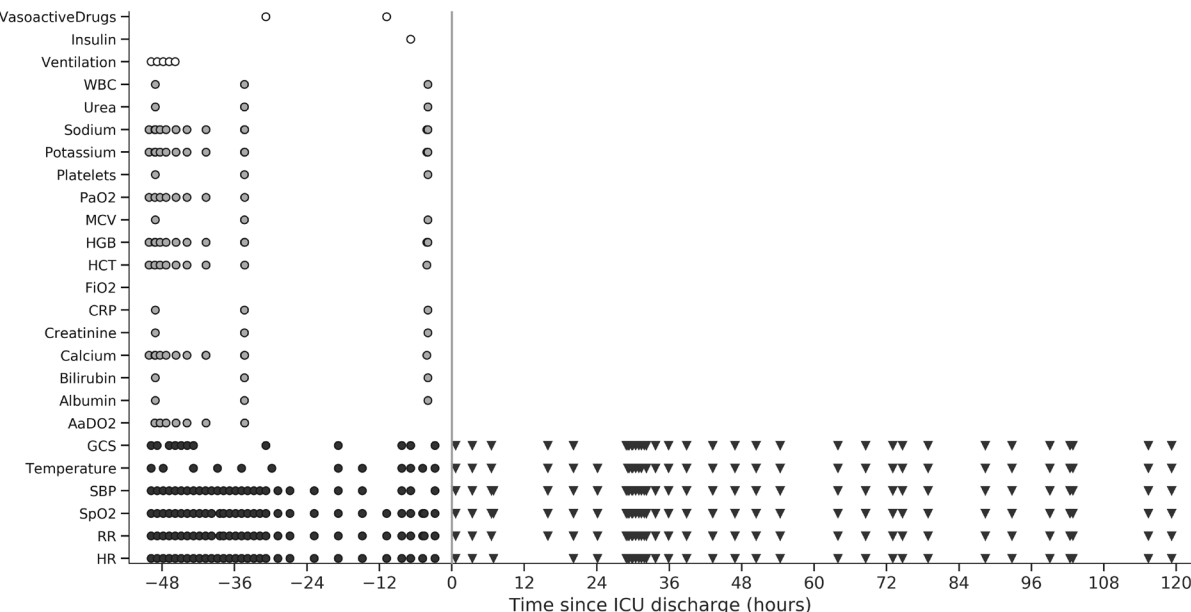

**Figure 2** Representation of a set of the variables acquired for an example patient included in our study pre-discharge (represented with circles) and post-discharge from the ICU. Variables include vital signs (dark grey), laboratory tests (lighter grey) and interventions/treatments (white) performed. The grey vertical line marks the patient's ICU discharge time point. A more detailed electronic patient record is 'generated' during the patient's ICU stay. We also note the different frequency of measurement of the vital signs in the ICU from that on the ward. $AaDO_2$, alveolar–arterial oxygen tension difference; CRP, C reactive protein; $FiO_2$, fractional inspired oxygen; GCS, Glasgow Coma Scale; HCT, haematocrit; HGB, haemoglobin; HR, heart rate; ICU, intensive care unit; MCV, mean corpuscular volume; $PaO_2$, arterial oxygen pressure; RR, respiratory rate; SBP, systolic blood pressure; $SpO_2$, peripheral oxygen saturation; WBC, white blood cell.

$$RSI = \beta \left[ \left(1 - \frac{t}{Tmax}\right) RS_1 \right] + \left[ \left(\frac{t}{Tmax}\right) RS_2 \right] \quad (1)$$

where $\beta$ is used to adjust the weight of $RS_1$ with respect to the time since discharge from ICU, $t$ corresponds to the elapsed time (in hours) since the patient was discharged from the ICU and has a maximum value of *Tmax* hours. Further details of the model development and optimisation of the parameters are available (see online supplemental file 1).

### Model validation and statistical analysis

The discrimination of the first model, $RS_1$, was assessed using the area under the receiver operating characteristic curve (AUROC) metric. Calibration was assessed using a goodness-of-fit test, the Hosmer-Lemeshow 'C' statistic, the Brier score and Cox's calibration regression.[25–27] The performance of this first model was examined both for the compound outcome and each adverse event (in-hospital death and ICU readmission) individually. The ability of $RS_1$ to predict future adverse events at increasing intervals from ICU discharge was also examined by calculating the AUROC for future events by day after (ICU) discharge (up to 120 days).

The final model (RSI) was validated using the AUROC for the compound outcome of in-hospital death or ICU readmission within the next N hours of a vital sign measurement recorded after ICU discharge, in line with previous studies for evaluating EWS systems.[20 21] We evaluated the model for different values of N, with N=(12, 24, 36, 48, 72) hours. We note that in this case, the AUROC represents how well the scoring system RSI discriminates between observation sets followed by an adverse event and those with no subsequent adverse outcome within the next N hours. Therefore, the unit of analysis is a vital sign set rather than a patient admission, as performed for the validation of the first model.

We also considered each individual adverse event separately. To understand better the feasibility of implementing the risk scoring systems in this setting, we also evaluated the burden of observation sets 'triggered' by the risk scoring system for every correctly identified observation followed by an adverse event within 24 hours.

We report the cross-validation results using the development dataset. We also report the external validation results using data from the RBH Trust. CIs were estimated using bootstrap CIs via percentiles, with 500 samples.[28]

### Comparison with published risk scoring systems

We compared the performance of our proposed RSI with that of each model individually ($RS_1$ and $RS_2$), and with that of a number of published and clinically used EWS

**Table 1** Details of the ICU admissions and the subgroups of post-ICU admissions included in the development and validation cohorts

| | Development (OUH) | Validation (RBH) |
|---|---|---|
| **ICU admissions included, n** | **8162** | **3421** |
| Age, years | 62 (47–72) (59) | 65 (51–75) (62) |
| Female sex, n | 3381 (41.4%) | 1480 (43.3%) |
| LoS in ICU, days | 1.9 (1.0–4.0) (4.2) | 2.0 (1.0–4.5) (4.2) |
| Outcome | | |
| ICU readmission, n | 543 (6.7%) | 154 (4.5%) |
| Time to ICU readmission, hours | 92 (37–205) (225) | 101 (43–283) (340) |
| In-hospital death, n | 462 (5.7%) | 218 (6.4%) |
| Time to in-hospital death, hours | 296 (84–781) (701) | 204 (66–600) (540) |
| Adverse event, n | 902 (11.1%) | 329 (9.6%) |
| **Post-ICU admissions included, n** | **273** | **136** |
| Age, year | 64 (51–72) (60) | 66 (55–73) (63) |
| Female sex, n | 85 (31.1) | 61 (45) |
| LoS in ICU, days | 1.9 (1.0–3.9) (3.5) | 1.3 (0.9–3.0) (3.4) |
| Outcome | | |
| ICU readmission, n | 18 (6.6%) | 9 (6.6%) |
| Time to ICU readmission, hours | 162 (73–220) (215) | 108 (72–202) (153) |
| In-hospital death, n | 6 (2.2%) | 4 (2.9%) |
| Time to in-hospital death, hours | 216 (66–514) (356) | 322 (244–365) (288) |
| Adverse event, n | 23 (8.4%) | 11 (8.1%) |

Continuous (numerical) variables are displayed as median (IQR) (mean), and count variables are displayed as counts (%). 'OUH' refers to admissions from the Oxford University Hospitals NHS Foundation Trust's two adult ICUs. 'RBH' refers to admissions from the Royal Berkshire NHS Foundation Trust's single adult ICU.
ICU, intensive care unit; LoS, length of stay.

systems: the modified EWS,[29] the standardised EWS,[30] the National EWS or NEWS,[21] and our centile-based EWS or CEWS.[31] We detail the components and weightings of the individual EWS systems in online supplemental file 2.

### Patient and public involvement
The PICRAM Study (2011–2015) had no embedded patient and public involvement in either conceptualisation or delivery of the work.

### RESULTS
The application of inclusion and exclusion criteria to patient admissions in order to derive the final cohorts in both organisations included in this study is shown in online supplemental figure SM3-1 (see online supplemental file 3). A total of 8162 admissions to the ICU were included in the development dataset. Vital sign observations were prospectively collected for 273 patients (3.3% from the 8162 ICU admissions) after discharge from the ICU. The validation dataset included 3421 ICU admissions, from which 136 (4.0%) had data acquired during the post-ICU period. Table 1 provides a summary of characteristics of the development and validation cohorts derived from the ICU admissions. Both cohorts were similar in terms of demographic and administrative data. The compound outcome rate was slightly higher in the development dataset than that in the validation dataset, considering either all ICU admissions or the subgroup of post-ICU admissions. We also note that the number of in-hospital deaths is higher than the number of readmissions to the ICU. In the development cohort, 12 394 vital sign observation sets were acquired from the 273 patients after ICU discharge, and 4831 from the 136 patients in the validation cohort.

For the first model ($RS_1$), which estimates the risk of future adverse events immediately after ICU discharge, 45 features from the 161 candidate features identified from the systematic review and expert opinion were retained in the final model. These are listed in online supplemental table SM3-1 (see online supplemental file 3). They comprise largely measures of cardiac or respiratory physiology, renal and hepatic function, plasma electrolytes, measures of inflammation and measures of treatment intensity. Of note, artificial ventilation during the last 24 hours of ICU admission

**Table 2** Performance of the first model (RS$_1$) on both development and validation datasets for the combined outcome (or adverse event) of in-hospital death or readmission to the ICU at any point after discharge from the ICU

| | Adverse event | In-hospital death | ICU readmission |
|---|---|---|---|
| **Development dataset** | | | |
| Hosmer-Lemeshow statistic | 50.81 | 339.07 | 224.94 |
| Brier score | 0.087 | 0.050 | 0.065 |
| Cox's calibration | | | |
| $\alpha$ | −0.009 | −0.037 | 0.013 |
| $\beta$ | 0.988 | 1.104 | 0.472 |
| AUROC (SD) | 0.782 (0.022) | 0.843 (0.022) | 0.716 (0.022) |
| **Validation dataset** | | | |
| Hosmer-Lemeshow statistic | 57.35 | 175.68 | 275.08 |
| Brier score | 0.081 | 0.058 | 0.058 |
| Cox's calibration | | | |
| $\alpha$ | 0.017 | −0.012 | 0.033 |
| $\beta$ | 0.627 | 0.475 | 0.268 |
| AUROC (SD) | 0.723 (0.014) | 0.780 (0.017) | 0.623 (0.019) |

Performance for each event separately is also displayed. Cox's calibration regression: for a good calibration, α should be close to 0, and β should be close to 1. AUROC is shown with mean (SD).
AUROC, area under the receiver operating characteristic curve; ICU, intensive care unit.

was associated with a lower risk of adverse outcome. This is because all the ICUs admit ventilated, elective, post-surgical admissions who have a low risk of adverse events and are discharged within 24 hours of extubation. Calibration plots of the model for both development and validation sets are shown in online supplemental figure SM3-2 (see online supplemental file 3). Table 2 summarises the performance for the combined outcome and for either adverse event (in-hospital death or ICU readmission) considered individually.

Table 3 shows the performance of RSI and other baseline scoring systems for predicting observation sets followed by in-hospital death or ICU readmission within the following 24 hours. The proposed scoring system showed an increased discrimination ability to predict adverse events within 24 hours with respect to the other risk scoring systems considered in this study. Using the external validation dataset (RBH), RSI gave an AUROC of 0.724 (95% CI of 0.704 to 0.741), vs 0.653 (0.621 to 0.683) for NEWS and 0.672 (0.648 to 0.695) for CEWS. Figure 3 shows the AUROC values of the risk scoring systems for predicting the compound outcome within 12, 24, 36, 48 and 72 hours of a vital sign measurement. The proposed RSI system consistently shows superior discrimination for each derived outcome.

## DISCUSSION
There are many variants of the original EWS system,[6–8] including systems designed for specific patient groups

such as children[32 33] and patients in high-dependency units.[34] These systems typically use vital signs to determine the level of risk of an adverse event. Vital signs are used because they are easily acquired variables, regularly measured in clinical practice and have a long history of being used to track patients' progression over time. However, point estimates of the risk of future deterioration from single vital sign measurement sets assume 'normal' values that may not be appropriate for a hospital population, and ignore other data that can add to the precision and granularity of the risk estimate. The introduction of electronic recording of vital signs and electronic patient records means that the parsimony and simplicity required by paper-based systems are now less relevant.

### Main findings and strengths
This was a proof-of-concept study for developing and validating an enhanced EWS that uses many more electronically held variables than the conventional vital signs, and which combined dynamic and static methods of risk evaluation, as typically used in other prediction disciplines such as meteorology[35] and imaging.[36] We chose to study post-ICU patients because they have a detailed electronic record generated during their ICU stay, a high adverse event rate after ICU discharge and because a significant body of literature exists on variables associated with adverse outcomes. However, the design principles we used in this study can be applied to any acute care patient group where sufficient data are available electronically.

**Table 3** Area under the receiver operating characteristic curve, SD and corresponding 95% CI for the developed Risk Scoring Index (RSI) and other competitive Early Warning Score (EWS) systems, using adverse event (readmission to the ICU or in-hospital death) within 24 hours of an observation set as the (compound) outcome

| | Adverse event | In-hospital death | ICU readmission |
|---|---|---|---|
| **Development dataset** | | | |
| MEWS | 0.749 (0.022) (0.734 to 0.764) | 0.752 (0.055) (0.717 to 0.789) | 0.747 (0.021) (0.733 to 0.761) |
| SEWS | 0.754 (0.019) (0.741 to 0.765) | 0.785 (0.050) (0.753 to 0.819) | 0.744 (0.022) (0.730 to 0.759) |
| NEWS | 0.757 (0.020) (0.742 to 0.771) | 0.889 (0.030) (0.867 to 0.912) | 0.737 (0.022) (0.721 to 0.753) |
| CEWS | 0.761 (0.021) (0.748 to 0.775) | 0.887 (0.029) (0.866 to 0.908) | 0.740 (0.022) (0.727 to 0.754) |
| $RS_2$ | 0.767 (0.019) (0.755 to 0.780) | 0.859 (0.041) (0.836 to 0.889) | 0.745 (0.020) (0.732 to 0.758) |
| **RSI** | **0.782 (0.018) (0.770 to 0.794)** | **0.931 (0.007) (0.926 to 0.936)** | **0.753 (0.017) (0.743 to 0.764)** |
| **Validation dataset** | | | |
| MEWS | 0.632 (0.032) (0.610 to 0.655) | 0.697 (0.054) (0.661 to 0.733) | 0.632 (0.033) (0.610 to 0.655) |
| SEWS | 0.678 (0.037) (0.653 to 0.703) | 0.678 (0.057) (0.642 to 0.717) | 0.681 (0.038) (0.657 to 0.706) |
| NEWS | 0.653 (0.044) (0.621 to 0.683) | 0.675 (0.063) (0.630 to 0.719) | 0.649 (0.043) (0.622 to 0.679) |
| CEWS | 0.672 (0.036) (0.648 to 0.695) | 0.685 (0.061) (0.643 to 0.728) | 0.670 (0.036) (0.645 to 0.596) |
| $RS_2$ | 0.693 (0.040) (0.669 to 0.718) | 0.681 (0.059) (0.644 to 0.722) | 0.693 (0.041) (0.666 to 0.721) |
| **RSI** | **0.724 (0.027) (0.704 to 0.741)** | **0.725 (0.045) (0.694 to 0.757)** | **0.722 (0.028) (0.702 to 0.741)** |

Results are shown for each outcome independently using both development and validation datasets.
CEWS, centile-based EWS; ICU, intensive care unit; MEWS, modified EWS; NEWS, National EWS; SEWS, standardised EWS.

As part of this work, we developed a new scoring system for predicting in-hospital mortality and readmission to the ICU from data collected during a patient's ICU stay. A systematic review in 2013[37] identified seven different published scoring systems that quantify the risk of mortality or readmission at varying intervals after ICU discharge. However, only two studies verified their systems with an external (independent) validation dataset.[38 39] They obtained AUROC values similar to those found in this study.

RSI is a dynamic system that computes an updated risk estimate every time a new vital sign is recorded. It combines this dynamic risk estimate derived from the routinely measured vital signs on the acute ward ($RS_2$) with a 'static' risk estimate that is computed immediately after discharge from the ICU using data from the ICU CIS ($RS_1$). As the performance of $RS_1$ is expected to worsen as the time from discharge from the ICU increases because patients' conditions change (see online supplemental figure SM3-2), we used a time-dependent function to reduce its contribution over time, with the patient's risk score becoming increasingly determined by vital signs-based $RS_2$. The combined risk estimates as given by RSI improved the predictive power of our scoring system when compared with $RS_2$.

In this study, a compound outcome of in-hospital mortality and ICU readmission was used as opposed to only using in-hospital mortality.[20] Patients discharged from the ICU with curative intent may die in hospital in spite of full, timely and appropriate care. They may also die because they do not receive the care they need in a timely fashion. Developing a model using in-hospital mortality as the sole outcome would limit the power of such a model, as appropriately treated survivors, with a readmission to the ICU, would not contribute to the adverse events. Hence, there would be a risk that the resulting model would predict inevitable, rather than preventable, deaths, as also noted by others.[40 41] Our main goal was to develop a system that would identify patients who were likely to respond to earlier intervention of higher-acuity care. While ICU readmission does not capture all the appropriately treated survivors, there is no other marker readily

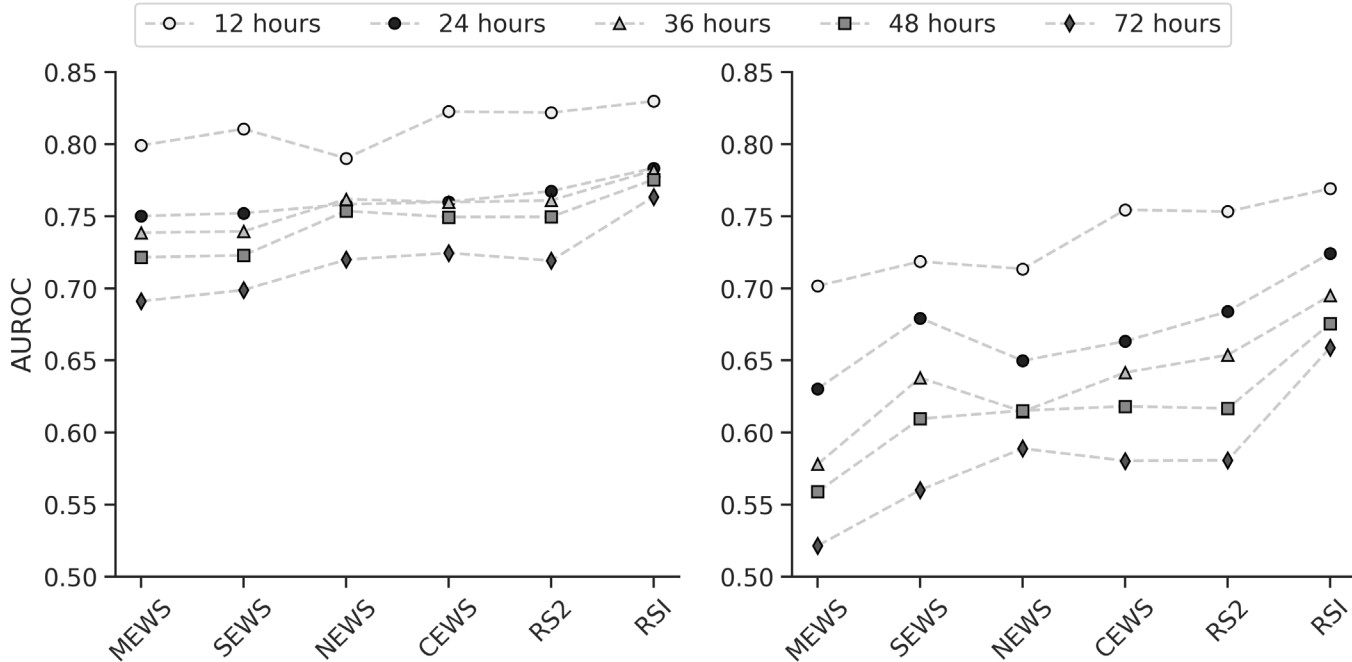

**Figure 3** Performance of the developed risk scoring system (RSI) and the other competitive scoring systems considered for predicting an adverse event (either readmission to the ICU or in-hospital death) within 12, 24, 36, 48 and 72 hours of a vital sign observation set. The left-hand panel shows the performance on the development dataset, while the right-hand panel shows the performance on the validation dataset. The performance is represented with the mean AUROC (area under the receiver operating characteristic curve) values. CEWS, centile-based Early Warning Score; ICU, intensive care unit; MEWS, modified Early Warning Score; NEWS, National Early Warning Score; RSI, Risk Scoring Index; SEWS, standardised Early Warning Score.

available in hospital electronic records to identify these patients. A similar compound outcome has been used in validation studies of EWS systems for the same reason.[21 31 42] We further note that RSI was considerably better at predicting in-hospital mortality than ICU readmission (see table 3). This has also been noted in other studies reporting evaluation of EWS systems,[21 42 43] where ICU admissions are less reliably identified than in-hospital death.

### Limitations
This work has a number of limitations. We studied a very specific group of patients who were at high risk of in-hospital deterioration and who had detailed records of their ICU stay before they were discharged to the ward. We could only study the variables available in the ICU electronic records; hence, some of the candidate variables identified in the systematic review could not be included in the model. In addition, as vital sign data had to be prospectively collected from consenting patients and transcribed from paper charts, the number of patients in the study, and therefore the number of adverse events, was limited.

We assessed the performance of the combined scoring system developed using the methodology commonly used to assess that of EWS systems. This uses AUROC measures based on paired vital sign recordings and events within fixed time periods; that is, derived outcome of occurrence of an adverse event

within, for example, 24 hours of a vital sign measurement. Therefore, AUROC values represent the probability that any randomly chosen observation followed within the chosen time period by in-hospital death or ICU readmission has a higher risk score than any randomly chosen observation not followed by an event in the same time period.[44] That is, repeated measurements from the same patient were used for evaluating the performance of scoring systems, which assumes that the scores computed from each observation set for that patient are independent (which is the usual approach when evaluating EWS systems). However, this assumption may not hold in practice; that is, a vital sign measurement at one point in time is likely to be correlated with previous measurements. This, alongside the highly imbalanced dataset (in which the outcome occurs infrequently), gives rise to AUROCs with high values that are not truly comparable with AUROCs from single-predictor/single-outcome algorithms where the unit of analysis is a patient admission.[45]

An external (independent) dataset was used to validate the risk scoring systems. RSI did not perform as well as expected for the combined outcome. This was primarily because the combined system overestimated the risk of adverse events in higher-risk patients, possibly reflecting a difference in the patient population at the two hospital sites.[46] The ICUs at the OUH admit tertiary referral patients not seen in RBH;

hence, risk associated with these patients captured in the scoring system built with the development dataset may not have added explanatory power for patient admissions in the validation dataset.

Finally, we note that the performance of our scoring system for predicting adverse events exceeds that of previously published EWS algorithms. This should, however, be interpreted with caution, as most of these systems were developed and validated on all hospital admissions to acute care areas, and our system was developed on a very specific population.

## CONCLUSIONS

Scoring systems, such as EWS systems, are used to identify hospitalised patients at risk of adverse events. In this study, we developed a bipartite score based on machine learning that encompasses the patient state at the time of ICU discharge, as well as vital signs recorded on the wards at the time the risk score is calculated. We showed that a scoring system incorporating data from a patient's stay in an ICU has better performance than typically used EWS systems based on vital signs alone.

**Acknowledgements** Dr Jonathan Burgess, who sadly passed away before seeing the fruits of his labour, undertook much of the early work on the identification of candidate variables in this project.

**Contributors** DAC—substantial contributions to the conception and design of the work, analysis and interpretation of data for the work, revising this manuscript critically for important intellectual content, final approval of the manuscript, agreement to be accountable for all aspects of the work. JLD—substantial contributions to the design of the work, revising this manuscript critically for important intellectual content, final approval of the manuscript, agreement to be accountable for all aspects of the work. AJ—substantial contributions to the acquisition and analysis of data for the work, revising this manuscript critically for important intellectual content, final approval of the manuscript, agreement to be accountable for all aspects of the work. MAFP—substantial contributions to the acquisition, analysis and interpretation of data for the work, revising this manuscript critically for important intellectual content, final approval of the manuscript, agreement to be accountable for all aspects of the work. IR—substantial contributions to the acquisition of data for the work, revising this manuscript critically for important intellectual content, final approval of the manuscript, agreement to be accountable for all aspects of the work. LT—substantial contributions to the conception and design of the work, substantial contributions to the acquisition, analysis and interpretation of data for the work, drafting the work and revising it critically for important intellectual content, final approval of the version to be published, agreement to be accountable for all aspects of the work. AW—substantial contributions to the acquisition of data for the work, revising this manuscript critically for important intellectual content, final approval of the manuscript, agreement to be accountable for all aspects of the work. PJW—substantial contributions to the conception and design of the work, substantial contributions to the acquisition, analysis and interpretation of data for the work, drafting the work and revising it critically for important intellectual content, final approval of the version to be published, agreement to be accountable for all aspects of the work. JDY—substantial contributions to the conception and design of the work, substantial contributions to the acquisition, analysis and interpretation of data for the work, drafting the work and revising it critically for important intellectual content, final approval of the version to be published, agreement to be accountable for all aspects of the work, and overall guarator for the work.

**Funding** This study was supported by the Department of Health and Wellcome Trust through the Health Innovation Challenge Fund (ref: HICF-0510-006).

**Disclaimer** The views expressed in this publication are those of the authors and not necessarily those of the Department of Health & Social Care or Wellcome Trust.

**Competing interests** DY, PJW and LT report grants from Wellcome Trust/UK Department of Health. PJW and LT report grants from the National Institute for Health Research and personal fees from Sensyne Health, outside the submitted work. PJW, DAC and LT are supported by the NIHR Biomedical Research Centre Oxford. MAFP is supported by a Drayson Research Fellowship.

**Patient and public involvement** Patients and/or the public were not involved in the design, or conduct, or reporting, or dissemination plans of this research.

**Patient consent for publication** Not applicable.

**Ethics approval** This study involves human participants. The project had UK Health Research Authority Confidentiality Advisory Group approval (ref: ECC 7-05(f)/2011), National Research Ethics Service Committee approval (ref: 11/SC/0440) for retrospective data collection and analysis, and National Research Ethics Service Committee approval (ref: 12/SC/0357) for prospective data collection and analysis. Participants gave informed consent to participate in the study before taking part.

**Provenance and peer review** Not commissioned; externally peer reviewed.

**Data availability statement** Data are available upon reasonable request. Fully anonymised individual participant data may be made available to researchers directly affiliated to an academic institution or charitable organisation who provide an independently reviewed, methodologically sound proposal. The data will be available for a period starting 6 months after publication and ending 1 year later. Only data required to answer the primary question posed by the proposed research will be made available. Researchers should contact Professor Young in the first instance. Other data and outputs from the project (Health Innovation Challenge Fund ref: HICF-0510-006) may be available on request.

**ORCID iDs**
Julie Lorraine Darbyshire http://orcid.org/0000-0002-7655-1963
Peter J Watkinson http://orcid.org/0000-0003-1023-3927

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
