## [Reviewer comments · BMJ Open]

ARTICLE DETAILS

TITLE (PROVISIONAL)	Development of an enhanced scoring system to predict ICU re-admission or in-hospital death within 24h using routine patient data from two NHS Foundation Trusts
AUTHORS	Pimentel, Marco; Johnson, Alistair; Darbyshire, Julie; Tarassenko, Lionel; Clifton, David A.; Walden, Andrew; Rechner, Ian; Watkinson, Peter; Young, Duncan

VERSION 1 – REVIEW

REVIEWER	Arnaud, Emilien University Hospital Centre Amiens-Picardie, Emergency Medicine
REVIEW RETURNED	05-Jul-2023

GENERAL COMMENTS	I thank the editor for this solicitation of reviewing, and especially the authors for this work. They defined a simple new model to predict readmission or death after ICU discharge based on routine clinical measures, and corrected by routinely clinical features collected in standard units. This model could be used with notmany effort in all hospitals. I find this study well performed and I think it should be published. I would propose some comments in the attached PDF
--

REVIEWER	Schuppert, Andreas RWTH Aachen University, Institute for Computational Biomedicine II
REVIEW RETURNED	03-Aug-2023

GENERAL COMMENTS	The paper addresses a pressing need in management of ICU patients. The authors provide highly valuable data, well selected for analysis of their research hypothesis. The analysis is appropriate, the mathematical parts of the study are clearly described such that a skilled reader should be able to reproduce the results. The results a quite clear and support the claim of the paper for utilisation of heterogeneous, multiple data structures in the design of clinically relevant predictors in order to improve their performance. Unfortunately the performance presented in the validation sets with ROC-AUC's just above 0.7 may be not sufficient for decision support systems, although it exceeds significantly the state of the art. As the authors write in the discussion this moderate performance may be due to the design of their study, namely to split training and validation sets according to the data source. It would be highly interesting if the authors could re-run their algorithm on a design splitting the
---

	training/test/validation set such that hospital bias is avoided in the splitting design. Anyway, the paper is definitely valuable to be published as it will be helpful for triggering and design of further analysis. Hence I would see the recommended add-on just as minor revision.
--	--

VERSION 1 – AUTHOR RESPONSE

Reviewer 1:

I thank the editor for this solicitation of reviewing, and especially the authors for this work. They defined a simple new model to predict readmission or death after ICU discharge based on routine clinical measures, and corrected by routinely clinical features collected in standard units. This model could be used with notmany effort in all hospitals. I find this study well performed and I think it should be published. I would propose some comments in the attached PDF	No specific revisions requested although we note that we did not receive an accompanying PDF. If there are recommendations in this document that we should address, please forward.
--	--

Reviewer 2:

The paper addresses a pressing need in management of ICU patients. The authors provide highly valuable data, well selected for analysis of their research hypothesis. The analysis is appropriate, the mathematical parts of the study are clearly described such that a skilled reader should be able to reproduce the results. The results a quite clear and support the claim of the paper for utilisation of heterogeneous, multiple data structures in the design of clinically relevant predictors in order to improve their performance. Unfortunately the performance presented in the validation sets with ROC-AUC's just above 0.7 may be not sufficient for decision support systems, although it exceeds significantly the state of the art. As the authors write in the discussion this moderate performance may be due to the design of their study, namely to split training and validation sets according to the data source. It would be highly interesting if the authors could re-run their algorithm on a design splitting the training/test/validation set such that hospital bias is avoided in the splitting design. Anyway, the paper is definitely valuable to be published	We agree with reviewer 2 that the proposed additional analyses would be interesting, and would enhance our evaluation of the algorithm. However, we regret that due to changes in personnel and professional roles in the interim years since the study ended we are unable to re-run the algorithm calculations as suggested.
---	---

as it will be helpful for triggering and design of further analysis. Hence I would see the recommended add-on just as minor revision.	
---	--

We thank both reviewers and the editorial team for their time and insightful comments when considering this submission. We hope our revisions at this stage are acceptable and we look forward to the next steps.

With kind regards

Julie Darbyshire, MA, MSc, DPhil
The University of Oxford
(Corresponding author)

VERSION 2 – REVIEW

REVIEWER	Arnaud, Emilien University Hospital Centre Amiens-Picardie, Emergency Medicine
REVIEW RETURNED	03-Nov-2023

GENERAL COMMENTS	Dear editor, dear authors, I thank you for the opportunity to assess the manuscript “Development of an enhanced scoring system to predict ICU re-admission or in-hospital death within 24h using routine patient data from two NHS Foundation Trusts”. As an emergency physician, I carefully reviewed the manuscript from the perspective of practitioner to understand how it could benefit patients, institutions and physicians in a daily practice. Unfortunately, the authors did not respond to the comments on the previous review in the annotated PDF uploaded in the appropriate ManuscriptCentral system. However, please find below all the previous comments restructured (and also in an uploaded PDF). 1 Major comments 1.1 Participants It is unclear why there are only 273 patients (from the 8 162 ICU admissions) included after the ICU discharged. It represents 3% of the total included ICU admissions (and the authors should have specified this percentage) The Figure SM3-1 should be included in the main manuscript. It should explain why 93% of post-ICU were not analyzed. The inclusion criterions must specify how the subset of post-ICU patients are selected (was it randomized ? why 273 patients ? was there any calculation of the necessary number of patients ?) In the result (P9L52) : The validation set contains 40% of inclusions but 50% of post-ICU vital signs. They seems to be randomized without post-ICU vital sign stratification consideration. It must be specified in the
---

	method. 1.2 Cohort comparison P9L53 “Both cohorts were similar in terms of demographic and administrative data”. The Table 1 should be statistically tested : is the validation set (from RBH) is comparable to the development set (from OUH) ? What ever is the result of this comparison, it should be discussed in the appropriate section. 2 Minor comments 2.1 Questions Some questions and clarifications are needed  – P6L20 The authors must clarify whether the data are automatically collected or manually collected during the second ward stay – P6L28 The authors should clarify why they chose 14 days – P6L52 The authors must specify the split ratio they used – Figure 1 : – Is FiO2 collected ? – Whats about non intubated patients, with low oxygen flow such as 12 L/min ? – Why SBP and not MBP which repreents in one measure SBP and DBP ? (does the delphi group justify this decision ?) 2.2 Need to reformulate  – P6L30 It is unclear why 55% of the data are manually double-entered. Do the authors want to say that the data were manually transfered from the observation paper to an electronic database manually, and for 55% of them, they were double-entered as a verification process ? – P6L55 Typoe “valida” – The flowchart (P9L47) is important and should be included in the main manuscript – P9L48 All counts of patients should be expressed in absolute number and in percentage of the total included patients. – Figure 1 : please include a legend within the chart 2.3 Need more references Some parts of the manuscript needs more references :  – P5L30 I hope these suggestions help you improve your manuscript. If you have any further questions or need more assistance, please feel free to ask. PS: The review has been written in English by myself and was subsequently copy-edited by ChatGPT for English correction.
--	--

VERSION 2 – AUTHOR RESPONSE

Reviewer Comment	Response
1.1 Participants It is unclear why there are only 273 patients (from the 8 162 ICU admissions) included after the ICU discharged. It represents 3% of the total included ICU admissions (and the authors should have specified this percentage) The Figure SM3-1 should be included in the main manuscript. It should explain why 93% of post-ICU were not analyzed. The inclusion criterions must specify how the subset of post-ICU patients are selected (was it randomized ? why 273 patients ? was there any calculation of the necessary number of patients ?) In the result (P9L52) : The validation set contains 40% of inclusions but 50% of post-ICU vital signs. They seems to be randomized without post-ICU vital sign stratification consideration. It must be specified in the method.	Data on patients treated in the ICU were collected (under ethics committee approval) without individual informed consent. The ethics approval required that individual informed consent was required for data to be collected from patients' post-ICU stay. 273 patients consented for this. We have clarified this in the manuscript, and added that this was 3.3% of those admitted to the ICU. Figure SM3-1 has been removed from the supplementary materials and a placement reference has been added to the main manuscript for a new Figure 1. We have clarified this in the text – the ICU datasets were retrospective, with permission to undertake research without patient consent. The post-ICU datasets are separately acquired from consented patients. We have clarified in the manuscript that a formal sample size calculation was not undertaken. As stated in the methods: After exclusion criteria were applied, we split the database into “development” and “validation” sets by organisation, such that (1) we developed and evaluated the model using data from all valid admissions within the OUH group using crossvalidation methods, and (2) we performed an external validation of the model using data from all valid admissions within the RBH group

1.2 Cohort comparison P9L53 “Both cohorts were similar in terms of demographic and administrative data”. The Table 1 should be statistically tested : is the validation set (from RBH) is comparable to the development set (from OUH) ? What ever is the result of this comparison, it should be discussed in the appropriate section.	There is no need or benefit in showing that the RBH and OUH cohort were statistically different from one another in any respect, so there was no comparison. It is sufficient that they were drawn from geographically distinct hospitals and patient population, from hospitals at different tiers in the UK healthcare system.
2.1 Questions Some questions and clarifications are needed  - P6L20 The authors must clarify whether the data are automatically collected or manually collected during the second ward stay - P6L28 The authors should clarify why they chose 14 days - P6L52 The authors must specify the split ratio they used - Figure 1 : Is FiO2 collected ? - Whats about non intubated patients, with low oxygen flow such as 12 L/min ? - Why SBP and not MBP which reprents in one 	P6L20 We have clarified in the main text that vital sign data were collected manually from patient records P6L28 – text added to clarify the decision to collect data for 14days after ICU discharge This timeline was chosen because the average duration of hospital stay after discharge from intensive care is ten days. P6L52 We now refer to the new figure 1 and specify in the text the number of patient admissions used in the development and validation models. Figure 1 (now figure 2): FiO2 was not collected for all patients. This is an example patient from the dataset, for whom FiO2 was not collected. Where FiO2 was collected for patients who were not intubated, this was estimated from oxygen flow rates When we started the project we did not know if blood pressure readings had any predictive value, and if they did whether this was the systolic pressure, diastolic pressure or both. We therefore retained the full detail of the blood pressure readings.

measure SBP and DBP ? (does the delphi group justify this decision ?)	
2.2 Need to reformulate - P6L30 It is unclear why 55% of the data are manually double-entered. Do the authors want to say that the data were manually transferred from the observation paper to an electronic database manually, and for 55% of them, they were double-entered as a verification process ? - P6L55 Type "valida" - The flowchart (P9L47) is important and should be included in the main manuscript	P6L30 – re-worded manuscript to clarify why 55% of patients' data were double-entered To verify the data entry process, 55% of patients' data were double entered. P6L55 – corrected to 'valid' Flowchart: Removed from supplementary pages and a placement reference added to the main manuscript for a new figure 1
- P9L48 All counts of patients should be expressed in absolute number and in percentage of the total included patients. - Figure 1 : please include a legend within the chart	P9L48 Clarified in the main text that 273 patients is 3.3% of 8163 ICU admissions and that 136 is 4% of 3,421 admissions The figure 1 (this is now figure 2 due to the transfer of SM3-1 to the main manuscript) legend was provided with the submission documentation. It is: Representation of a set of the variables acquired for an example patient included in our study pre (represented with circles) and post discharge from the ICU. Variables include vital signs (dark grey), laboratory tests (lighter grey) and interventions/treatments

	(white) performed. The grey vertical line marks the patient's ICU discharge timepoint. A more detailed electronic patient record is "generated" during the patient's ICU stay. We also note the different frequency of measurement of the vital signs in the ICU from that on the ward, which would be expected as frequency of observations declines with decreasing severity of illness.
2.3 Need more references Some parts of the manuscript needs more references : – P5L30	We believe the manuscript is generally adequately referenced without taking it to excess. P5L30 is a part of materials and methods that does not require referencing. Apologies if we have missed something.

VERSION 3 – REVIEW

REVIEWER	Arnaud, Emilien University Hospital Centre Amiens-Picardie, Emergency Medicine
REVIEW RETURNED	01-Mar-2024
GENERAL COMMENTS	Dear authors, Thank you for your corrections or the explanations to my questions. All my concerns ar addressed.